# Mathematical Modeling of Induction Heating of Waveguide Path Assemblies during Induction Soldering

**Vadim Tynchenko** [1,2,3,*], **Sergei Kurashkin** [1,2], **Valeriya Tynchenko** [1,2], **Vladimir Bukhtoyarov** [1,2], **Vladislav Kukartsev** [1,2], **Roman Sergienko** [4], **Viktor Kukartsev** [1] and **Kirill Bashmur** [1]

[1] Department of Technological Machines and Equipment of Oil and Gas Complex, School of Petroleum and Natural Gas Engineering, Siberian Federal University, 660041 Krasnoyarsk, Russia; scorpion_ser@mail.ru (S.K.); 051301@mail.ru (V.T.); vladber@list.ru (V.B.); vlad_saa_2000@mail.ru (V.K.); vakukartsev@sfu-kras.ru (V.K.); bashmur@bk.ru (K.B.)
[2] Information-Control Systems Department, Institute of Computer Science and Telecommunications, Reshetnev Siberian State University of Science and Technology, 660037 Krasnoyarsk, Russia
[3] Marine Hydrophysical Institute, Russian Academy of Sciences, 299011 Sevastopol, Russia
[4] Gini Gmbh, 80339 Munich, Germany; roman@gini.net
[*] Correspondence: vadimond@mail.ru; Tel.: +7-95-0973-0264

**Abstract:** The waveguides used in spacecraft antenna feeders are often assembled using external couplers or flanges subject to further welding or soldering. Making permanent joints by means of induction heating has proven to be the best solution in this context. However, several physical phenomena observed in the heating zone complicate any effort to control the process of making a permanent joint by induction heating; these phenomena include flux evaporation and changes in the emissivity of the material. These processes make it difficult to measure the temperature of the heating zone by means of contactless temperature sensors. Meanwhile, contact sensors are not an option due to the high requirements regarding surface quality. Besides, such sensors take a large amount of time and human involvement to install. Thus, it is a relevant undertaking to develop mathematical models for each waveguide assembly component as well as for the entire waveguide assembly. The proposed mathematical models have been tested by experiments in kind, which have shown a great degree of consistency between model-derived estimates and experimental data. The paper also shows how to use the proposed models to test and calibrate the process of making an aluminum-alloy rectangular tube flange waveguide by induction soldering. The Russian software, SimInTech, was used in this research as the modeling environment. The approach proposed herein can significantly lower the labor and material costs of calibrating and testing the process of the induction soldering of waveguides, whether the goal is to adjust the existing process or to implement a new configuration that uses different dimensions or materials.

**Keywords:** induction heating; mathematical modeling; process; control; automation; optimization; waveguide

## 1. Introduction

Modeling the process of induction soldering is one of the easier and more effective ways to improve the quality of process control, which helps to enhance the ultimate product. The authors in [1–3], describe how such modeling can be used for quality improvement in the photovoltaic industry. The authors of [1] present a multiphysical model of induction soldering for making modular solar panel systems. The model presented in the paper can be used to optimize the process of induction soldering by adjusting the geometry of the head to heat the solder more efficiently and evenly (the "head" hereafter refers to the work-head of the induction soldering unit). The model also makes adjustments for deviations resulting from the specifics of materials in use.

The authors in [2,3] use modeling to analyze the effects of cell cracking on the performance of solar cells. Their simulations show that cracks do not necessarily compromise

the performance of photovoltaic modules. The authors in [4] present an induction heating model implemented in Cedrat Flux 10.3, a commercial package. Experimental tests prove the model to be accurate. The simulation results are consistent with the experimental temperature profile of the specified surface points. The model has the advantage of being able to predict such parameters as current density and magnetic flux field inside a workpiece—these parameters are difficult to measure directly.

The authors in [5] show that reduced-order models constitute a fairly effective and promising tool for controlling the induction soldering process by means of indirect measurements. A fourth-order system is obtained by proper orthogonal decomposition. The model does enable the process to attain the desired temperature; at the same time, it is sufficiently simple from the computational standpoint to run on a relatively cheap microcontroller.

As part of the research presented in [6], the authors developed a thermal model of the infrared soldering process. The model is able to predict thermal effects from the parameters of convection in an infrared furnace to the detailed thermal response including the solid–liquid transition of the solder. The authors in [7–9] present a neural-network model of induction soldering process control. The model greatly improves the quality of controlling the process of induction soldering for spacecraft waveguides. The authors in [10] describe the use of a neural-network model developed to control the process of oil-and-gas equipment repair and maintenance.

The authors of [11,12] present a mathematical model of the process of spacecraft waveguide induction soldering. These papers clearly demonstrate the high quality of the proposed models. The proposed models have also been proven to be quite efficient for this application.

The authors in [13] present a mathematical model, which is a neural-fuzzy controller for the process of waveguide induction soldering. Experimental tests in kind prove the model to be accurate.

The simulation modeling of the soldering process is the topic of many works. Satheesh et al. [14] carry out the numerical estimation of soldering process in terms of strain fields and localized transient temperature. Works [15–17] are devoted to the issues of modeling and analysis of induced stresses, as well as the effect of the soldered joint geometry on such parameter. The authors of [18] propose a mathematical model of the process of induction brazing of products made of nonmagnetic metals, considering the existing nonlinearities of the process under consideration and using the orthogonal expansion to reduce the complexity of the numerical study of the proposed model. Given how commonly simulation modeling is used to optimize induction heating-based processes in a variety of mechanical engineering applications, it is clear that the approach is highly efficient when it comes to modeling such processes [19–21]. The author in [22] uses COMSOL Multiphysics simulation system to investigate how the head parameters affect the efficiency of heating parts and the maximum heating zone. The electromagnetic field parameters are modeled there at 22 to 100 kHz. The simulation results show that heating is fastest at lower frequencies; as such, electromagnetic fields penetrate deeper into the material of the parts.

However, in the works available in the public domain at the moment, the issues of modeling the induction heating process of special structures that are widely used in communication systems are not considered. The present work is intended to close the existing knowledge gap and to present a mathematical apparatus that provides high-quality modeling of the process of waveguide path soldering using induction heating.

The first steps of simulating the induction soldering process can be conveniently done in ready-made modeling software such as Simulink (a MATLAB extension) [23], which can be used for modeling both static [24] and dynamic systems [25]. Further, it is possible to verify the created models both in the COMSOL Multiphysics environment [26–29] and in the well-known ANSYS system [30–33]. As an alternative to the Simulink package, the present work uses the Russian software SimInTech [34–36]. SimInTech allows not only to

implement models of complex dynamic systems [37], but also to test systems for automated control of complex objects and technological processes [38].

In this work, the resulting mathematical models were used in the development of an adaptive (smart) process control system for the induction soldering of waveguide assemblies in order to obtain an even heating of soldered elements in order to make a high-quality permanent joint. The applicability and usability of the proposed algorithm is tested both by computational and in-kind experiments.

## 2. Materials and Methods

### 2.1. Mathematics Behind Models

In order to model the induction heating process of waveguides, it was necessary to develop mathematical models for individual waveguide assembly components and for entire waveguide assemblies. A waveguide assembly consists of a waveguide rectangular tube and a flange or a coupler.

As mathematical models for heating assembly elements, which are used to work out the technological process of the induction soldering of thin-walled aluminum waveguide paths, we write the expression of the temperature field with a continuously operating stationary source (1):

$$T(x,t) = \int_0^t \frac{Q}{c\rho F\sqrt{4\pi at}} e^{\left(-\frac{x^2}{4at} - bt\right)} dt \tag{1}$$

$$b = \frac{\alpha p}{c\rho F} \tag{2}$$

where $Q$ is the amount of heat [J], $F$ is the cross-section of a waveguide element [m$^2$], $x$ is the distance from the heat source [m], $cp$ is the volumetric heat capacity [J/m$^3$], $t$ is time [s], $b$ is the coefficient of convective heat transfer from the rod surface to the environment (see Equation (2)), $a$ is the thermal conductivity coefficient and $p$ is the cross-sectional perimeter.

The cross-section of an element of the waveguide path can be represented in the following form (3):

$$F = AB - A'B' \tag{3}$$

where $A$ and $B$ are the length and width of the product, respectively, and $A'$ and $B'$ are the length and width of the inner hole in the product, respectively.

For the tube, the assumption is made that the cross-section is heated evenly. Accordingly, an expression can be formulated for calculating the temperature field from the action of an instantaneous heating source in a flat rod, considering the input of geometric constraints (reflections), which is applicable for bodies bounded by mutually perpendicular planes (a parallel tubed, rectangular plate), for a rod of finite length and for an infinite wedge with an opening angle $\pi/n$, where n is an integer [39].

The essence of the reflection method is reduced to the expansion of the bounded body along the corresponding coordinate to infinity and the selection of additional sources in the extended area so that the boundary conditions on the surface of the bounded body are satisfied [39].

Based on the fact that the waveguide tube is limited on two sides, one side situated in the immediate vicinity of the induction heating source is as follows (4):

$$T(x,t) = \int\limits_{i=-\infty}^{\infty} \int\limits_{j=-1,1} \frac{Q}{c\rho F\sqrt{4\pi at}} e^{\left(-\frac{(x-jl-2iL)^2}{4at} - bt\right)} \tag{4}$$

where $x$ is the distance from the left side [m], $t$ is time [s], $a$ is the thermal conductivity coefficient, $L$ is the rod length [m], $l$ is the end-to-heat source distance and $j$ is the number of reflections taken into account in the calculation, selected in such a way that, for $j + 1$ for any $x$ and $t$, $T(x,t) \leq \varepsilon$ is valid as $\varepsilon \to 0$.

Equation (4) rapidly converges, and only a few terms are considered for real bodies. As the temperature is equalized, the number of retained members of the series increases [39]. An illustration of the computational domain is presented in Figure 1 describing the distribution of heat sources in the heated element of the waveguide path (tube/flange/coupling).

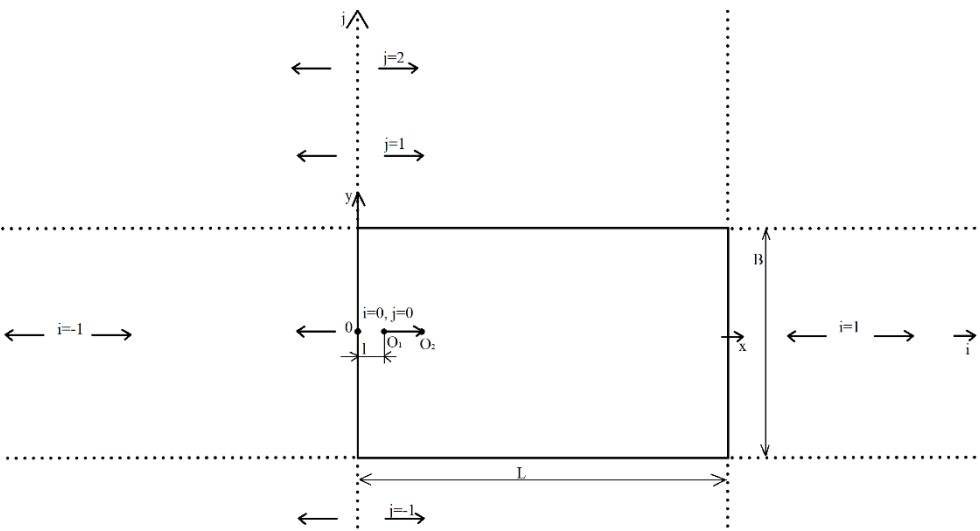

**Figure 1.** Heat source distribution scheme.

The diagram illustrates the movement of the energy source and point $O_1$ to point $O_2$ along the $x$-axis in time $t$. In this case, this source is reflected relative to the planes $x = 0$, $x = L$ and $y = B/2$ and $y = -B/2$.

Equation (4) makes it possible to represent the new mathematical models describing the heating of elements and assemblies of waveguide paths, which differ from the known ones considering the geometry of the products and allow the development of the parameters of the induction soldering process.

*2.2. Waveguide Assembly: Tube Model*

For a waveguide tube, the following assumptions must be made:

1. The tube is a sufficiently long body of a homogeneous material;
2. The cross-section of the waveguide tube along the entire length is constant;
3. The tube is similar to the rod in terms of heat transfer and thermal conductivity.

The conclusion here is that a mathematical model of a planar heat source in a rod holds true for a planar heat source in the rectangular tube of a waveguide assembly. This effectively introduces a geometric constraint on one side of the rod, whereas we denote and take into account the finiteness of the tube where the flange overhangs it when a respective joint is made. Another assumption is that the waveguide tube heats evenly over its cross-section as a result of being thin (<2 mm) and that the head is designed in a way to enable the even heating of the tube along its perimeter.

A realistic image of a standard waveguide assembly tube is shown in Figure 2. With the introduction of restrictions and the projection of the tube (Figure 3), the calculation in Equation (5) for the process of heating the waveguide tube, considering the geometric dimensions of the product, can therefore be written as follows:

$$T(x,t) = \sum_{j=-1,1} \frac{Q}{Fc\rho\sqrt{4at}} e^{\left(-\frac{(x+jl)^2}{4at} - bt\right)} \tag{5}$$

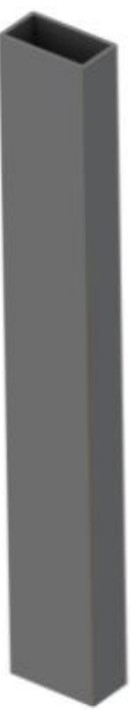

**Figure 2.** Photorealistic image of a waveguide assembly tube.

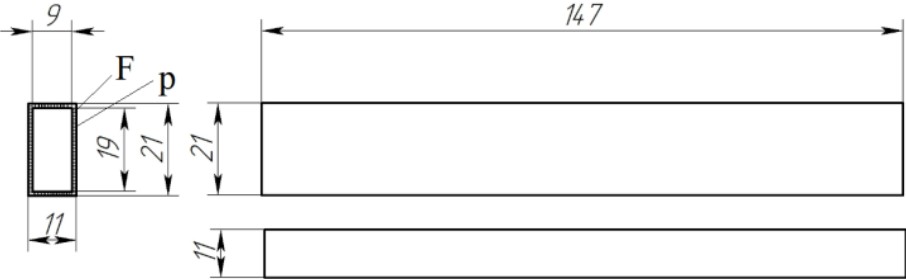

**Figure 3.** Projections of a waveguide assembly tube with its dimensions. F is the cross-sectional area of the tube; p is the cross-sectional perimeter.

To test the effectiveness of the proposed approach, the curves produced by the waveguide assembly tube heating model for various power levels of the induction heater are obtained (Figure 4). Thus, this calculation procedure is sufficiently consistent with the process of induction heating for waveguide assembly tubes.

*2.3. Waveguide Assembly: Flange Model*

Flange (Figure 5) heating is simulated in a similar manner to the methodology described above. A flange is a relatively small body, which means heat is withdrawn from it at a relatively low rate. This means that, at certain heating rates, the temperature will be uniform along the sectional axis near the soldering zone, which in turn means that no adjustment for temperature distribution in that plane is necessary. However, a flange is quite thin along its other axis, and therefore its heat distribution along that axis cannot be even. Unlike a tube, a flange is finite on either side, which means that the heat reflection off the finite boundaries of the body needs to be adjusted for.

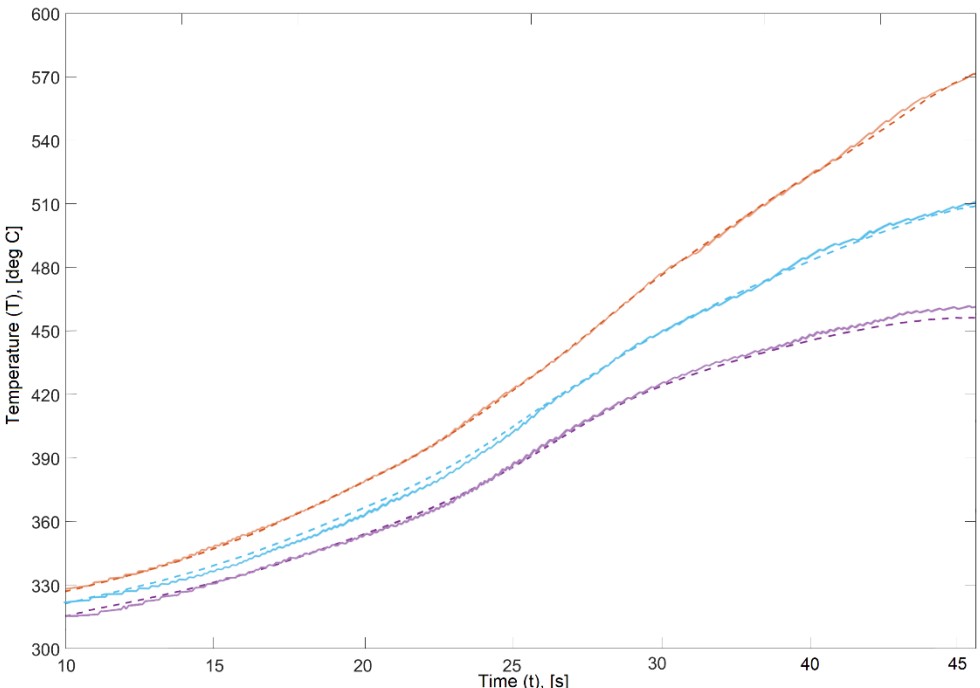

**Figure 4.** Curves produced by the waveguide assembly tube heating model, where dotted lines are model-derived data, solid lines are experimental data, orange curve is for 11 kW heating, blue curve is for 5 kW heating, and lilac curve is for 3 kW heating.

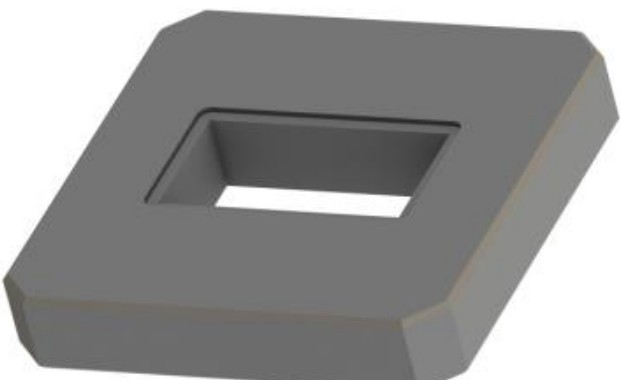

**Figure 5.** Photorealistic image of a flange.

Thus, with the introduction of constraints and the projection of the flange/coupling (Figure 6), in this study, a new calculation in Equation (6) for the heating process of the flange/coupling of the waveguide assembly is proposed:

$$T(x,t) = \sum_{i=0}^{j} \frac{2Q}{Fc\rho\sqrt{4at}} e^{\left(-\frac{(x+2iL)^2}{4at} - bt\right)} \tag{6}$$

where $x$ is the distance from the left side [m] and $L$ is the flange/coupler length [m].

The curves produced by the waveguide assembly flange heating model for various power levels of the induction heater are obtained (Figure 7). Thus, this calculation procedure is sufficiently consistent with the process of induction heating for waveguide assembly flanges.

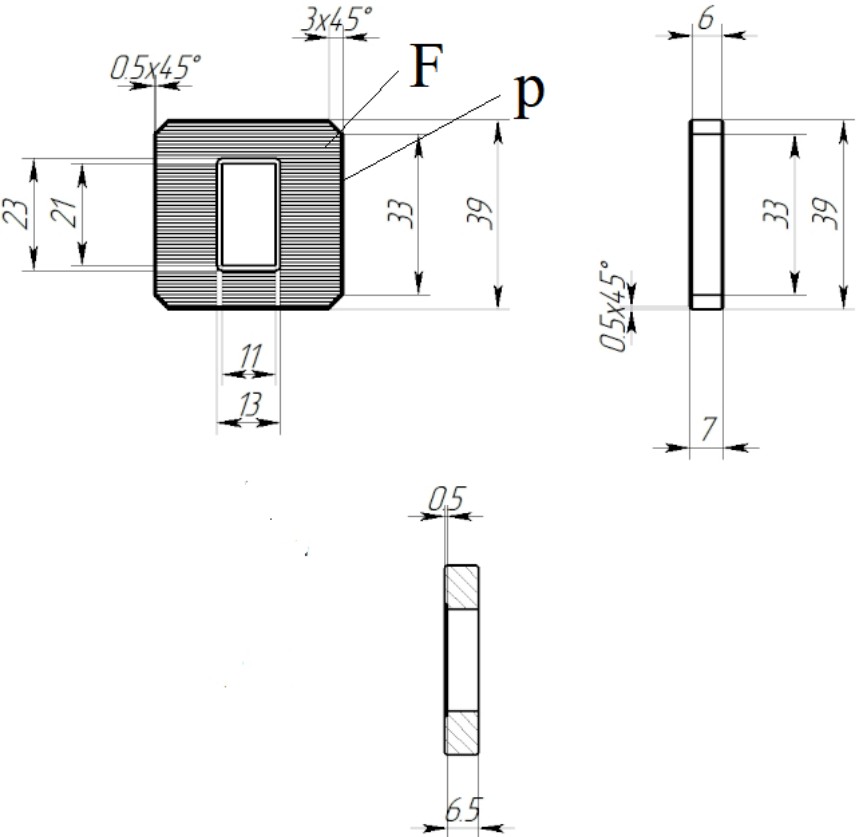

**Figure 6.** Projections of a waveguide assembly flange with its dimensions. F is the cross-sectional area of the flange; p is the cross-sectional perimeter.

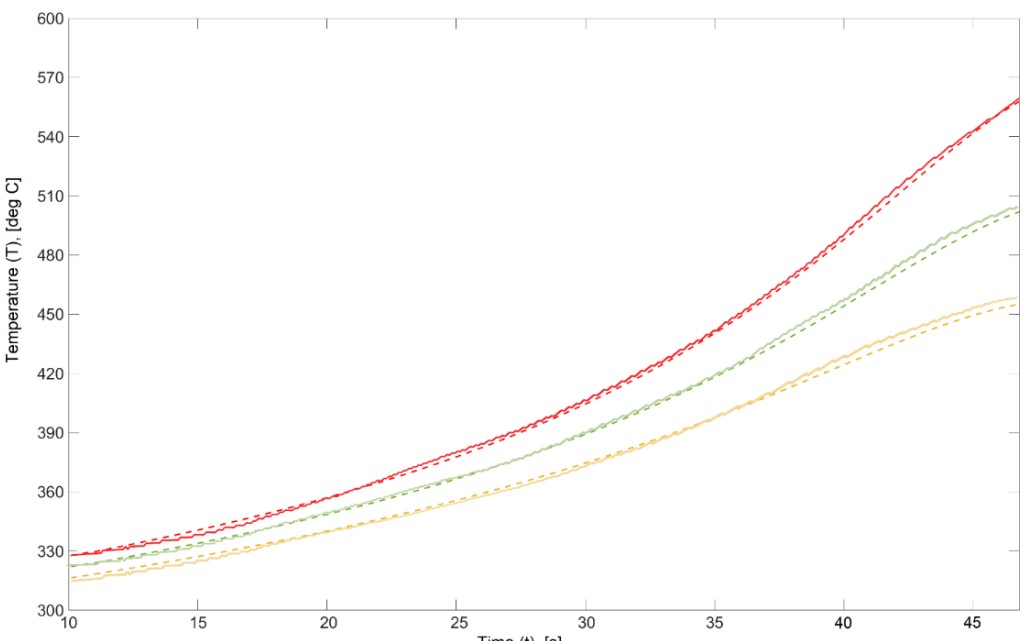

**Figure 7.** Curves produced by the waveguide assembly flange heating model, where dotted lines are model-derived data, solid lines are experimental data, red curve is for 11 kW heating, green curve is for 5 kW heating, and yellow curve is for 3 kW heating.

### 2.4. Waveguide Assembly Model

When modeling the distribution of heat energy between assembly components (Figure 8), the assumption is that there is a certain law of energy distribution that is bound to both the dimensions and the configuration of the head.

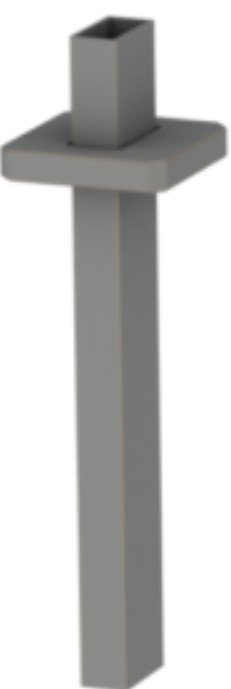

**Figure 8.** Standard tube–flange waveguide assembly.

The induction soldering of waveguides requires a head with a slanted opening to localize peak heat near the soldering zone. Given this specific feature of the technology, it can be assumed for modeling that all the head-transferred energy is released in the soldering zone. Thus, let us assume for convenience that all of the generator's energy will be transferred to the assembly being soldered. Therefore, the law of heat distribution between the waveguide assembly components will be as follows:

$$q(t) = q(t)M(x) + q(x)(1 - M(x)) \tag{7}$$

where $q(t)$ is a permanent heat source, $M(x)$ is the coefficient of heat distribution between the assembly components, $M(x)$ belongs to $[0, \ldots, 1]$ and $x$ belongs to $[n, \ldots, m]$, where $n$ and $m$ are, respectively, the upper boundary and the lower boundary, and $x$ is the flange/coupler to head distance.

For this research, the distribution function (7) was derived empirically. A design diagram (Figure 9) could explain the essence of the energy distribution during the heating of the waveguide array.

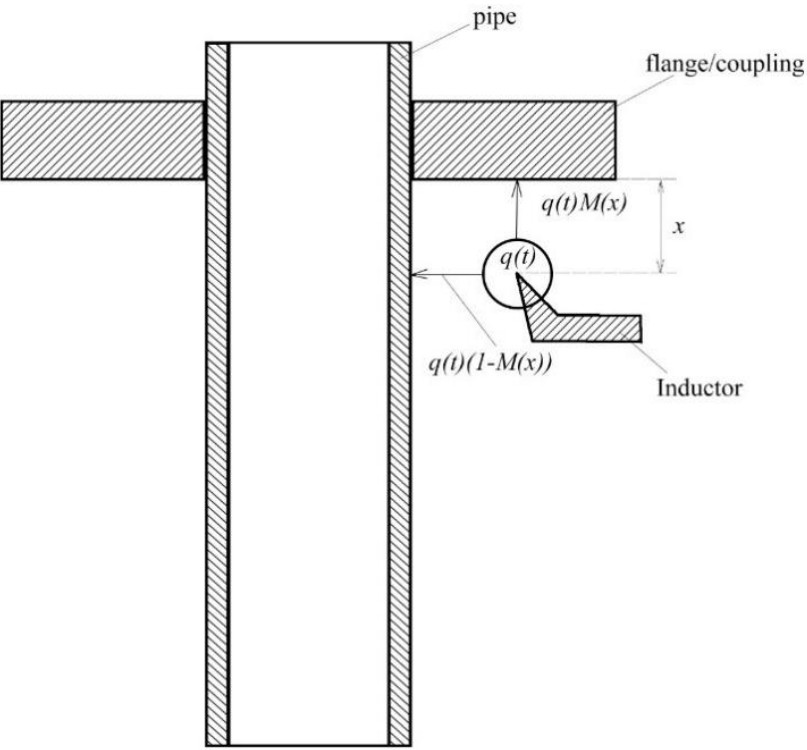

**Figure 9.** Diagram of the heat distribution of a continuously operating source of the waveguide assembly.

### 3. Experimental Study and Discussion

*3.1. Verification of Proposed Mathematical Models*

For the verification of the compliance of the developed mathematical models with the real technological process of the induction heating of waveguide paths, a series of experiments was conducted using an experimental installation for the induction soldering of waveguide paths for spacecraft, which includes a high-frequency generator (66 kHz), a voltage source (up to 10 V, with an output current of up to 150 A and a power of up to 15 kW), a matching device, a flat inductor with a working window of a 26 mm × 15 mm rectangular section and a manipulator–positioner.

Non-contact temperature sensors AST A250 are used to measure the temperature in the system. These pyrometers have the following characteristics: spectral range is 1.6 μm, accuracy is ±0.3% of the measured value + 1°C, distance to spot size ratio is 200:1 (350 °C–1800 °C). The temperature is measured from the flat surface of the flange in the area of the proposed connection with the pipe. On the pipe, measurements were carried out at a distance of 1 cm from its side in the area of maximum energy application formed by the magnetic field of the inductor. The layout of pyrometric sensors is presented in previous work [40].

The experiments were conducted for a tube and a flange for a waveguide path with a standard size of 22 mm × 11 mm by heating them and fixing the temperature values by means of contactless pyrometry. Summary graphs (Figure 10) could be used to compare the model-generated curves and real-world process curves of induction heating; the graphs are specific to the assembly components of spacecraft waveguides.

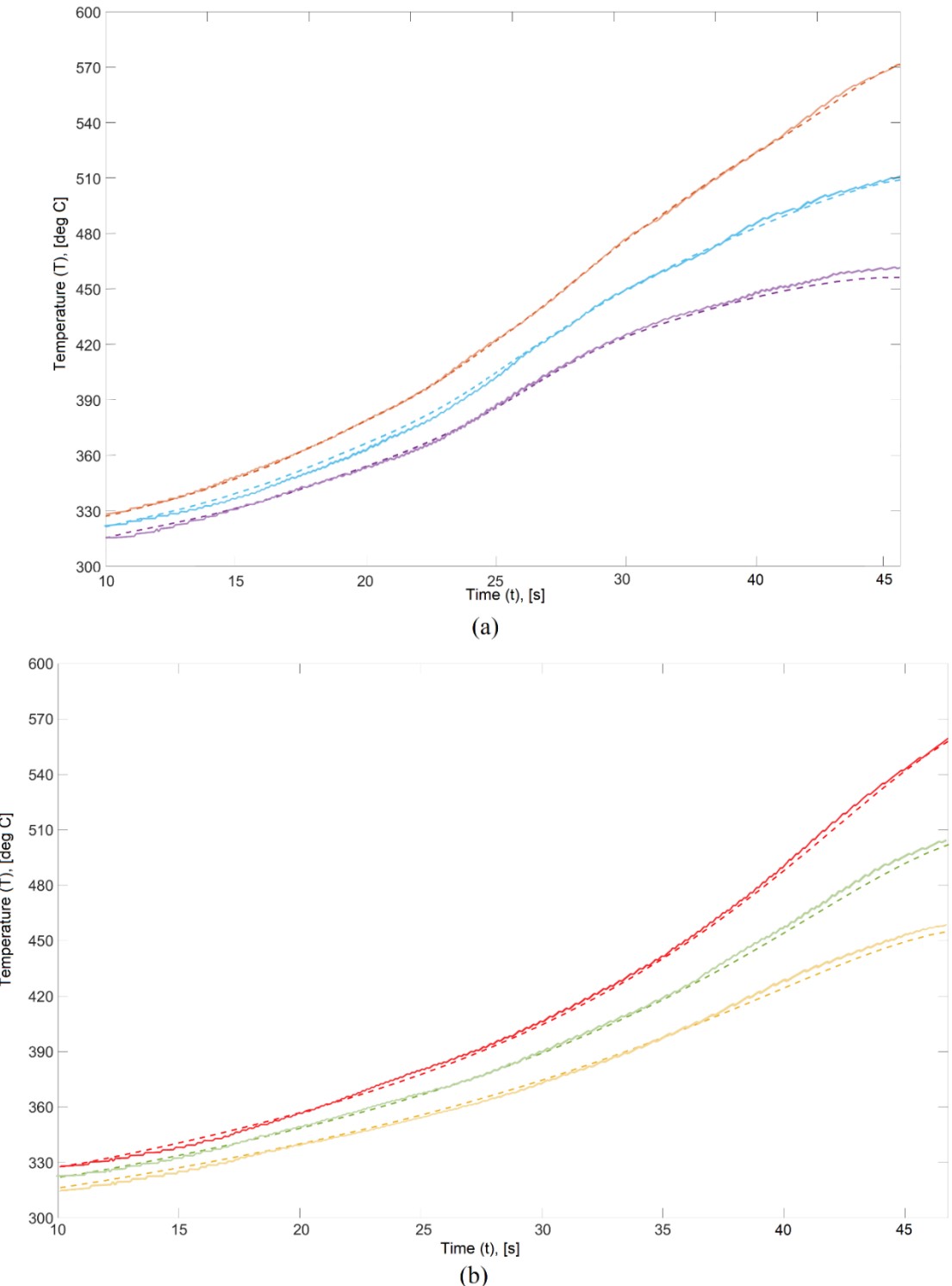

**Figure 10.** Waveguide component heating curves: (**a**) 22 mm × 11 mm tubes; (**b**) 22 mm × 11 mm flanges, where dotted lines are model-derived data, solid lines are experimental data, orange and red curves are for 11 kW heating, blue and green curves are for 5 kW heating, and lilac and yellow curves are for 3 kW heating.

From the graphs shown in Figure 10, it can be seen that the mathematical models proposed in this work repeat the real technological process of heating the elements of the waveguide path with a high degree of reliability. In this case, the waveguide tube experiences more intense heating in comparison with the flange, which is due to the small thickness of the tube.

As can be seen in Figure 10 as well as in the standard error values given in Table 1, the induction heating models developed for thin-walled aluminum waveguide assemblies were proven to be accurate in simulating this process. In-kind and simulation experiments showed that the models could indeed be used to test and calibrate the process parameters for the induction soldering of thin-walled aluminum spacecraft waveguides.

**Table 1.** Standard deviation: simulation vs. real-world processes.

| Waveguide Component | Heating Power P, kW | | |
|---|---|---|---|
| | 3 | 5 | 10 |
| Tube | 1.4 | 1.6 | 1.5 |
| Flange | 1.5 | 1.8 | 1.7 |

### 3.2. Implementation of the Proposed Models to Calibrate the Induction Soldering Control Algorithms

To test and calibrate the process of making a permanent joint, it is necessary to implement a model in the simulation system of choice, which in this case was SimInTech [34,35].

SimInTech (Simulation in Technic) [36] is an environment for the dynamic simulation of technical systems; it is designed to validate process control systems for complex facilities. SimInTech simulates processes found in various industries and can also model control systems to help improve the design of such systems by testing every decision at any stage of the project [37]. SimInTech is designed to investigate and analyze in detail nonstationary processes in nuclear and thermal power plants, automatic control systems, tracking drives and robots or any other systems whose dynamics can be described by a system of differential algebraic equations and/or implemented by structural modeling methods. SimInTech is mainly intended for making models, designing control algorithms and debugging them on the object model, as well as for generating source codes in C for programmable controllers [38].

The model the authors implemented in SimInTech was an automated control system that comprised a control action, actuators, the controlled object and feedback (Figure 11).

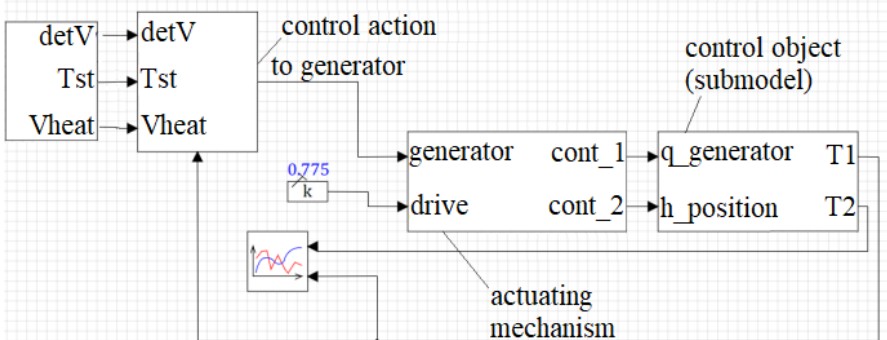

**Figure 11.** Overview of the SimInTech model, where detV is the deviation in heating rates, Tst is the stabilization temperature, Vheat is the workpiece heating rate, generator is the actuator input, q_generator is the control action, T1 is the waveguide tube temperature and T2 is the flange/coupler temperature.

The induction soldering of thin-walled waveguides was implemented as a proportional-integral-differentiating (PID) controller (Figure 12), which uses the set of actuators (Figure 13). The amplification factor is marked with a blue number in the Figure 13 and at the given moment is equal to 1. The upper circuit is the transfer function of a standard generator; the lower circuit is the transfer function of a standard feedback-enabled motor. The deconstructed controlled object in SimInTech shown in Figure 14 is of particular interest.

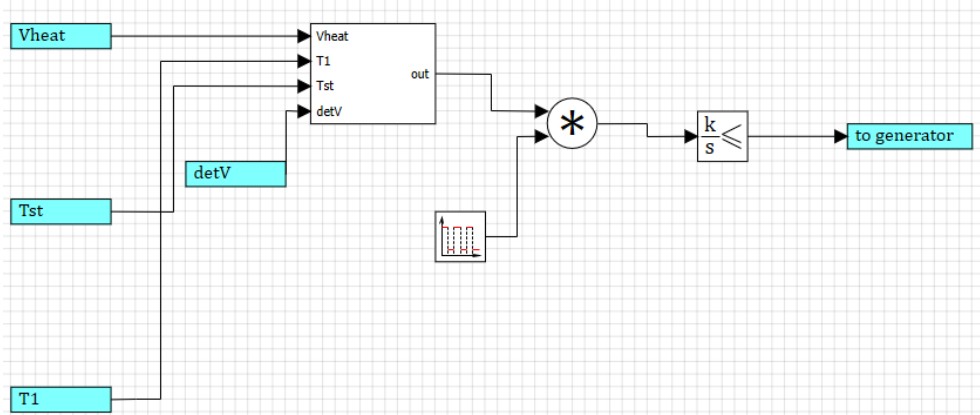

**Figure 12.** Deconstruction of the control action block in SimInTech, where detV is the deviation in heating rates, Tst is the stabilization temperature, Vheat is the workpiece heating rate, generator is the actuator input, q_generator is the control action, T1 is the waveguide tube temperature, * is the multiplier, k/s is the integrating link and to_generator is the control action output.

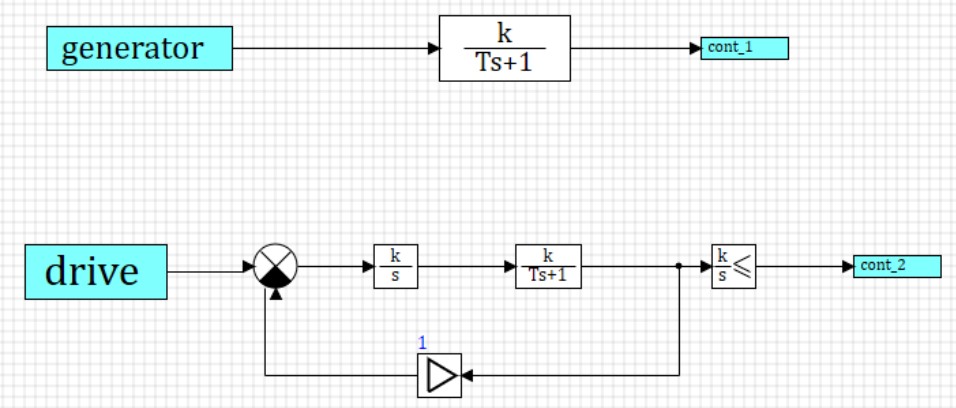

**Figure 13.** Deconstruction of actuators in SimInTech, where generator is the actuator input, drive is the input of the positioner, cont_1 is the control action for heating and cont_2 is the control action for workpiece positioning.

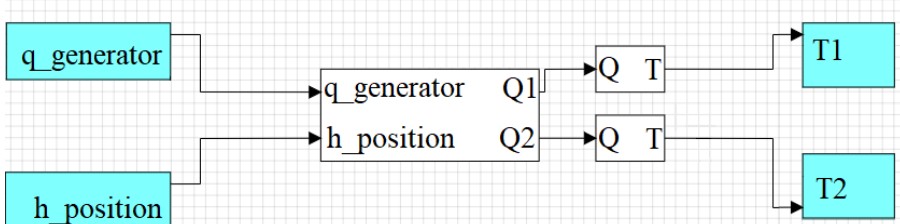

**Figure 14.** Deconstruction of the controlled object in SimInTech.

This submodel centers around a block that simulates the distribution of heat between the waveguide assembly components. It has the following inputs: q_generator and h_position. The block is followed by blocks that implement the waveguide assembly components: the upper one is used for a tube and the lower one for a flange. The outputs are current temperatures of the corresponding waveguide assembly components.

Curves of the control process for soldering a 22 × 11 tube–flange assembly plotted by SimInTech is shown in Figure 15. The head to assembly distance was set to 4 mm. The assembly heating rate was altered, as the process was controlled by the flange temperature: 18 °C/s (Figure 15a), 14 °C/s (Figure 15b) and 10 °C/s (Figure 15c).

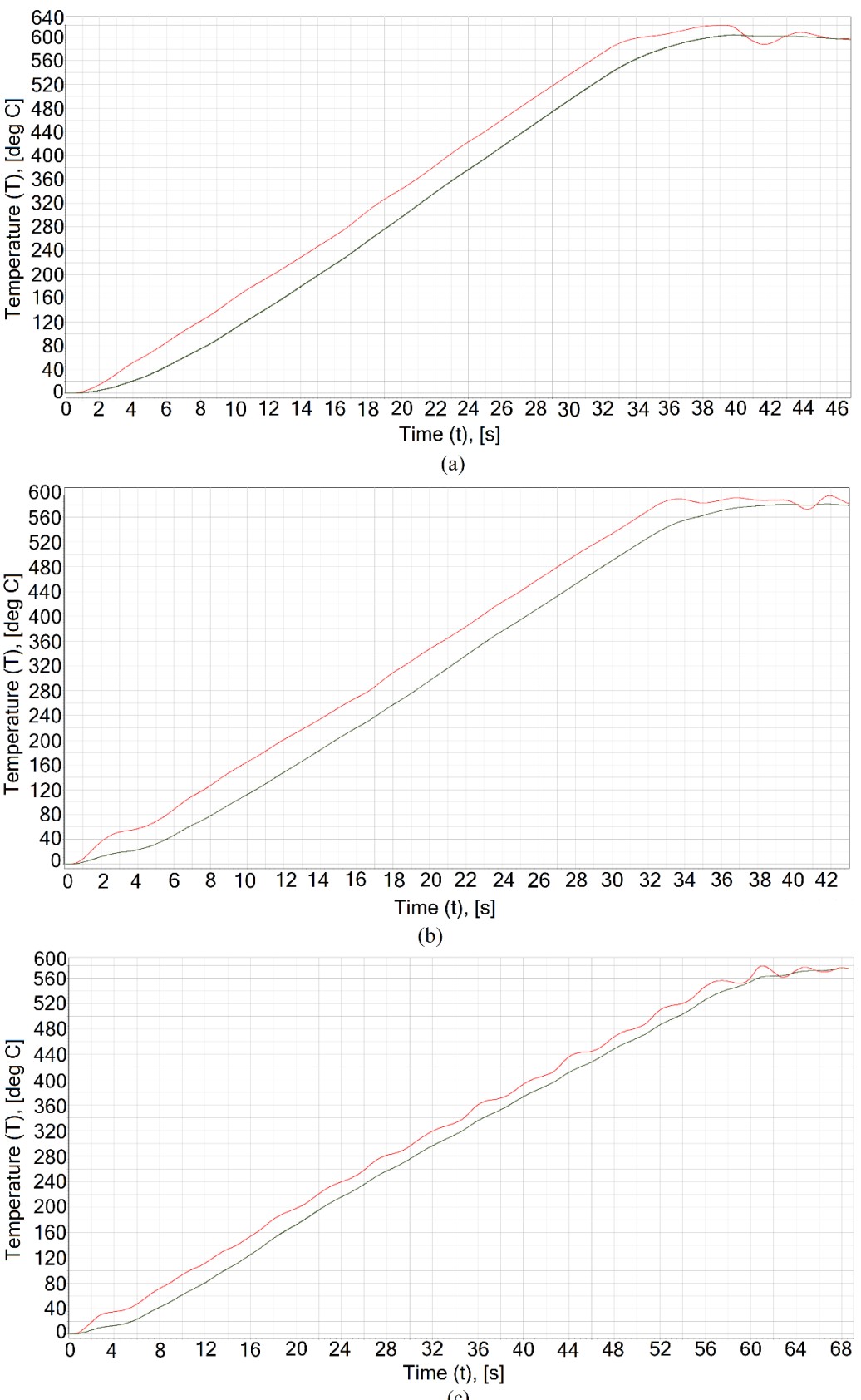

**Figure 15.** Waveguide induction soldering process control curves, with a distance of 4 mm to the inductor window: (**a**) 18 °C/s, (**b**) 14 °C/s, (**c**) 10 °C/s. The red curve is for the waveguide tube temperature ($T^{Tb}$), and the green curve is for the flange temperature ($T^{Fl}$).

The simulation experiment showed that the temperatures of the soldered components had the least scatter at 10 °C/s. Simulations at a lower heating rate were considered unadvisable due to the peculiarities of the soldering technology, as flux would only be active for a limited time.

To eliminate the temperature scatter, soldering was again simulated at 10 °C/s but, this time, with a 3 mm distance between the waveguide assembly and the inductor window; this redistributed temperatures from the waveguide tube to the flange. Figure 16 shows the simulation results that prove the process to be of high quality.

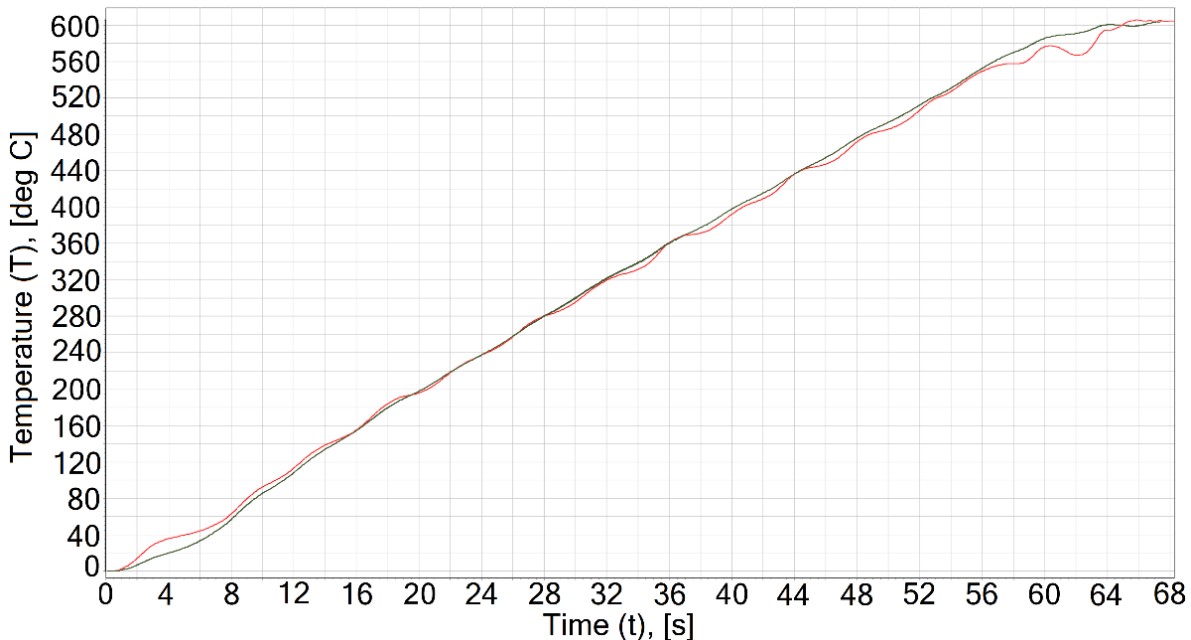

**Figure 16.** Waveguide inductions' soldering process control curves, with a distance of 3 mm to the inductor window, 10 °C/s: the red curve is for the waveguide tube temperature ($T^{Tb}$), and the green curve is for the flange temperature ($T^{Fl}$).

In order to exclude the spread of temperatures of the elements to be soldered, the soldering process was simulated at a rate of 10 °C/s, but the distance from the waveguide assembly to the inductor window was reduced to 3 mm for the temperature redistribution from the waveguide tube on the flange. The simulation results (Figure 16) show a good fit of the model to the real process.

Table 2 summarizes the numerical values of the operation quality of the automatic control system of induction soldering in the process of optimizing the control parameters: the heating rate of the waveguide assembly and the distance between the flange/coupling and the inductor window. The standard deviation is evaluated by the next Equation (8):

$$SD = \sqrt{\frac{1}{n}\sum_{i=1}^{n}\left(T_i^{Diff} - \overline{T}^{Diff}\right)^2} \tag{8}$$

where $T_i^{Diff} = \left|T_i^{Tb} - T_i^{Fl}\right|$ is the absolute waveguide elements temperature difference, $\overline{T}^{Diff} = \frac{1}{n}\sum_{i=1}^{n}\left|T_i^{Tb} - T_i^{Fl}\right|$ is the mean value of waveguide elements temperature difference, $n$ is the number of temperature control points, $T_i^{Tb}$ is the tube temperature in $i$-th point, $T_i^{FL}$ is the flange temperature in $i$-th point.

**Table 2.** Overregulation (OR) and standard deviation (SD): flange/coupling temperature vs. tube temperature (in °C).

| Distance from the Waveguide Assembly to the Inductor Window, mm | Heating Rate, °C/s | | | | | |
|---|---|---|---|---|---|---|
| | 18 | | 14 | | 10 | |
| | OR | SD | OR | SD | OR | SD |
| 3 | - | - | - | - | 0.8 | 4.3 |
| 4 | 18.2 | 39.7 | 10.8 | 38.4 | 4.8 | 21.1 |

A multitude of experiments can be run to test soldering parameters on various waveguide assembly configurations by using the mathematical models developed here for testing and calibrating the induction soldering process for thin-walled aluminum waveguides. The use of the models can reduce R&D costs as they enable pre-calibrated experimentation in kind, which effectively cuts the costs of consumables.

## 4. Conclusions

The goal of this work was to develop mathematical models of induction soldering for waveguide assembly components, which could help in testing and calibrating the induction soldering process for the thin-walled aluminum waveguides found in spacecraft. To verify the developed models, the research team carried out simulations and tests in kind, which showed the models were indeed accurate in predicting the actual induction heating processes observed in the induction soldering of thin-walled aluminum waveguides found in spacecraft.

The standard deviation of the temperature difference between simulation and real-world processes did not exceed 2 °C, which is satisfactory in terms of induction soldering technology.

In addition, the authors studied the application of the developed mathematical apparatus for calibrating the control of the induction soldering process. The results showed that the found effective values of the heating rate and the distance from the wave-guide assembly to the inductor window allowed the control of the technological process with low overregulation (0.8 °C) and for a low standard deviation between the temperatures of the soldered elements to be maintained (4.3 °C).

In future, we aim to extend this research to achieve the following:

- To develop a set of adaptive induction soldering process control methods that employ state-of-the-art data mining algorithms;
- To implement a prototype spacecraft waveguide induction soldering control system based on the developed mathematical models and algorithms;
- To test the applicability, usability and effectiveness of such a prototype system with computational and in-kind experiments;
- To design a process diagram to integrate the proposed software prototype into an existing experimental system for the induction soldering of spacecraft waveguides.

Integrating the developed models in the existing process control hardware and software for spacecraft waveguide induction soldering will help to cut costs due to the use of the models to test and calibrate different induction soldering parameters on a variety of waveguide configurations and sizes.

**Author Contributions:** Conceptualization, V.T. (Vadim Tynchenko); Data curation, V.B. and V.K. (Viktor Kukartsev); Formal analysis, S.K., V.T. (Valeriya Tynchenko), V.B., V.K. (Vladislav Kukartsev) and R.S.; Investigation, V.T. (Vadim Tynchenko), S.K., V.B. and V.K. (Viktor Kukartsev); Methodology, V.K. (Vladislav Kukartsev) and R.S.; Project administration, V.T. (Vadim Tynchenko); Resources, V.K. (Vladislav Kukartsev) and K.B.; Software, V.T. (Valeriya Tynchenko), V.B. and V.K. (Vladislav Kukartsev); Supervision, V.T. (Vadim Tynchenko); Validation, V.T. (Vadim Tynchenko) and V.T. (Valeriya Tynchenko); Visualization, V.T. (Vadim Tynchenko), S.K. and K.B.; Writing—original draft, V.T. (Vadim Tynchenko), S.K., V.T. (Valeriya Tynchenko), V.B., V.K. (Vladislav Kukartsev), R.S., V.K. (Viktor Kukartsev) and K.B.; Writing—review and editing, V.T. (Vadim Tynchenko), S.K., V.T. (Valeriya Tynchenko), V.B., V.K. (Vladislav Kukartsev), R.S., V.K. (Viktor Kukartsev) and K.B. All authors have read and agreed to the published version of the manuscript.

**Funding:** This research received no external funding.

**Institutional Review Board Statement:** Not applicable.

**Informed Consent Statement:** Not applicable.

**Conflicts of Interest:** The authors declare no conflict of interest.

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
