# Peer review of "Mathematical Modeling of Induction Heating of Waveguide Path Assemblies during Induction Soldering"

_metals, doi:10.3390/met11050697_

Round 1
Reviewer 1 Report
the manuscript could be accepted.
Author Response
Dear Reviewer 2,
Thank you for your positive evaluation of our work.
As for English quality, we have used MDPI English Editing service to improve English language of our paper (English-Editing-Certificate-28437).
With best regards,
Dr. Vadim Tynchenko

Reviewer 2 Report
The paper 'Mathematical Modeling of Induction Heating of Waveguide Path Assemblies during Induction Soldering' presents a research on induction heating modeling using analytical expressions. The developed model is applied and verified using an experimental setup. In addition, the results obtained using a commercial software are provided. The authors demonstrate the agreement between the analytical an experimental data. However, several point related to the research presentation and methodology require enhancement prior to further consideration. Here I provide my remarks.
- The first paragraph and one part of the abstract are completely the same. Such feature should be avoided.
- The part of the Introduction between lines 83 and 95 includes lumped references. Please, if possible avoid rfs lumping and provide at leas one sentence for each of the cited work.
- Line 96 - delete extra space between 'models' and 'were'.
- The end of the Introduction section should provide more details on the contribution of the paper in the context of the literature review. A gap analysis should be stressed out more clearly.
- Units are usually written in [ ] brackets, lines 112-119.
- Line 136: 'the left end' - a more general qualification, related to Fig. 1 i suggested. Left side if a relatively loosely definition.
- The authors usually use expression 'Formula (number)' throughout the paper. A substitute term like equation, Eq., of expression is suggested.
- Generally, references to figures are kept as stand alone paragraphs throughout the manuscript. A more reader friendly integration of those sentences within the rest of the text body is strongly advised.
- I would suggest tu use term rectangular tube instead of 'pipe'. A pipe is usually used in the case of circular cross sections conduction some kind of media.
- The statement in the line 212 seems out of the context. It is not clear why do the author now mention different assembly combination.
- Experimental study: please provide additional data on the temperature measurement technique and equipment.
- Line 256 - is it Fig 10 instead of Fig 8? Line 261 - which figure?
- Please provide ref regarding the used software in lines 269-281.
- Fig 11 requires editing regarding font size. I is not clear why do the author avoid usage of indices and write, e.g. detV instead of detV.
- Please, provide the applied expression related to the standard deviation evaluation.
- Line 298 - this kind of 'solo paragraphs' should be avoided.
- Line 317 - Figure 15 - please reconsider the relation of each figure with the flange temperature once again. Present data is inconsistent.
- Figs 15 and 16 - a green line can barely bee seen. Please correct.
- A clear comparison between experimental data, analytical modeling and the data obtained using the SimInTech software is missing to a great extent. Presently (Section 3.2) I could not capture the clearly elaborated relation between these three approaches.
- Please, provide the prospects for the future research within the concluding section.
- Editing issues: delete extra tabulator in lines like 121, 136, 197, etc.
- Grammar issues, lines: 103, 133-134, 288
Author Response
Dear Reviewer 3,
Thank you for your positive comments on our paper. You provided remarks about it, so we’ve made changes in the paper.
Remark 1: The first paragraph and one part of the abstract are completely the same. Such feature should be avoided.
Reply: Thank you for the remark. We have changed the introduction to be different from the abstract.
Remark 2: The part of the Introduction between lines 83 and 95 includes lumped references. Please, if possible avoid rfs lumping and provide at leas one sentence for each of the cited work.
Reply: Thank you for the comment. We have now changed the Introduction, so it almost has no lumped references (except when several works are considered by us in aggregate). Lines 72-87, 94-101.
Remark 3: Line 96 - delete extra space between 'models' and 'were'.
Reply: Thank you for the comment. We have made correction (line 102).
Remark 4: The end of the Introduction section should provide more details on the contribution of the paper in the context of the literature review. A gap analysis should be stressed out more clearly.
Reply: Thank you for the comment. We have now added the paragraph about it (lines 88-93).
Remark 5: Units are usually written in [ ] brackets, lines 112-119.
Reply: Thank you for the comment. We have made corrections throughout the paper (lines 118-120, 141-142, 201).
Remark 6: Line 136: 'the left end' - a more general qualification, related to Fig. 1 i suggested. Left side if a relatively loosely definition.
Reply: Thank you for the comment. We have made correction, so now it sounds ‘left side’ (lines 141, 201).
Remark 7: The authors usually use expression 'Formula (number)' throughout the paper. A substitute term like equation, Eq., of expression is suggested.
Reply: Thank you for the comment. We have changed ‘Formula’ to ‘Eq.’ everywhere in the paper.
Remark 8: Generally, references to figures are kept as stand alone paragraphs throughout the manuscript. A more reader friendly integration of those sentences within the rest of the text body is strongly advised.
Reply: Thank you for the comment. We have made corrections throughout the paper.
Remark 9: I would suggest tu use term rectangular tube instead of 'pipe'. A pipe is usually used in the case of circular cross sections conduction some kind of media.
Reply: Thank you for the comment. We have changed term ‘pipe’ with ‘tube’ throughout the paper.
Remark 10: The statement in the line 212 seems out of the context. It is not clear why do the author now mention different assembly combination.
Reply: Thank you for the comment. We have changed this paragraph (lines 212-214).
Remark 11: Experimental study: please provide additional data on the temperature measurement technique and equipment.
Reply: Thank you for the comment. We have provided additional data on the temperature measurement technique and equipment (lines 244-251).
Remark 12: Line 256 - is it Fig 10 instead of Fig 8? Line 261 - which figure?
Reply: Thank you for the comment. We have changed the Figure number (line 257) and added the number to Figure in line 262.
Remark 13: Please provide ref regarding the used software in lines 269-281.
Reply: Thank you for the comment. We have now added refs in lines 279-290.
Remark 14: Fig 11 requires editing regarding font size. I is not clear why do the author avoid usage of indices and write, e.g. detV instead of detV.
Reply: Thank you for the comment. We have now changed the font sizes in Figure 11. As of avoiding usage of indices: the program system SimInTech is not allows using indices. So, we had to use usual text style in our model.
Remark 15: Please, provide the applied expression related to the standard deviation evaluation.
Reply: Thank you for the comment. We have now added the equation for standard deviation evaluation (lines 349-355).
Remark 16: Line 298 - this kind of 'solo paragraphs' should be avoided.
Reply: Thank you for the comment. We have now changed all the paper text to avoid 'solo paragraphs'.
Remark 17: Line 317 - Figure 15 - please reconsider the relation of each figure with the flange temperature once again. Present data is inconsistent.
Reply: Thank you for the comment. We’ve made a mistake in the paper text. We have now changed the temperature rates in line 324.
Remark 18: Figs 15 and 16 - a green line can barely bee seen. Please correct.
Reply: Thank you for the comment. We have now changed the green color in Figures 15 and 16 to make it more visible.
Remark 19: A clear comparison between experimental data, analytical modeling and the data obtained using the SimInTech software is missing to a great extent. Presently (Section 3.2) I could not capture the clearly elaborated relation between these three approaches.
Reply: Thank you for the comment. The Section 3.2 is devoted to the practical implementation of our mathematical heating models. Experimental data is used to verify the proposed mathematical models in Section 3.1.
Due to the fact that the pyrometric sensors (that we use) have a measurement interval from 300 to 1800 degrees, all comparisons of the results of mathematical modeling in Section 3.1 are made in the range from 300 degrees to 570 (the melting temperature of the aluminum alloy used in the production of waveguide paths).
Section 3.2 already discusses the practical application of the models. In the SimInTech system, we simulate a control system for the induction soldering process and show that the proposed models allow not only assessing the temperature distribution in the heating zone of waveguides, but also selecting effective process control algorithms. For this, graphs are presented that show the temperatures of the soldered elements of the waveguide path in the control process. All the data in Section 3.1 is model data.
Remark 20: Please, provide the prospects for the future research within the concluding section.
Reply: Thank you for the comment. Within the framework of this work, the results of a completed study on modeling the energy distribution during induction heating of waveguide paths are presented. The proposed models make it possible, to a sufficient extent for practical use, to estimate the temperature distribution in such specialized products. Our authors team sees further areas of research in the sense of using such models for practical purposes - to improve the quality of soldered joints in the automation systems of the induction soldering process. We tried to describe this in lines 380-392.
Remark 21: Editing issues: delete extra tabulator in lines like 121, 136, 197, etc.
Reply: Thank you for the comment. We have now changed all the paper text to delete extra tabulators after Equations.
Remark 22: Grammar issues, lines: 103, 133-134, 288.
Reply: Thank you for the comment. We have changed the paper text (lines 109, 138-139, 299-300).
With best regards,
Dr. Vadim Tynchenko

Round 2
Reviewer 2 Report
I would like to thank to the Editorial office for an opportunity to review a paper. Also, I thank to the authors for careful considerations of my remarks.
This manuscript is a resubmission of an earlier submission. The following is a list of the peer review reports and author responses from that submission.
Round 1
Reviewer 1 Report
This study developed mathematical models of induction soldering for 313 waveguide assembly components by carrying out simulations and experiments to help test and calibrate the induction soldering pro-314 cess of thin-walled aluminum waveguides in spacecraft. The research is meaningful and the results are positive. However, the format and font of the picture should be adjusted to make it easier to read. And English writing needs improvement.
Reviewer 2 Report
1. Most of the images are poor processed and used in the manuscript.
2. the conclusions are not well supported by the datas.
Author Response
Reviewer 2:
1. Most of the images are poor processed and used in the manuscript.
R: Thank you for the comment. We have now adjusted the format and font of the pictures to make it easier to read.
2. the conclusions are not well supported by the datas.
R: Thank you for the comment. We have now added the data to “3. Experimental Study and Discussion” and “4. Conclusions” parts to support the conclusions.
Reviewer 3 Report
This work presents a mathematical modeling of induction heating to test and calibrate process control methods for induction soldering of waveguides. Up to the reviewer’s opinion, the focus of this manuscript is out of the scope of Metals and, therefore, it must be rejected. The authors are encouraged to look for a journal with focus on process control.